# UNIVERSAL JAILBREAK BACKDOORS IN LARGE LANGUAGE MODEL ALIGNMENT

**Thomas Baumann**
ETH Zurich
`thomas.baumann@inf.ethz.ch`

## ABSTRACT

Aligning large language models is essential to obtain models that generate helpful and harmless responses. However, it has been shown that these models are prone to jailbreaking attacks by reverting them to their unaligned state via adversarial prompt engineering or poisoning of the alignment process. Prior work has introduced a "universal jailbreak backdoor" attack, in which an attacker poisons the training data used for reinforcement learning from human feedback (RLHF). This work further explores the universal jailbreak backdoor attack, by applying it to other alignment techniques, namely direct preference optimization (DPO), identity preference optimization (IPO), Kahneman-Tversky optimization (KTO) and odds ratio preference optimization (ORPO). We compare our findings with previous results and question the robustness of the named algorithms.

## 1 INTRODUCTION

Large language models (LLM) are pretrained on vast corpora such as the common webcrawl (Schäfer, 2016). To be usable as a chatbot or assistant they need to undergo further tuning, usually *instruction tuning* and *alignment*. In instruction tuning a model gets trained to follow various orders given in natural language, resulting in a model that can generalize well to unfamiliar tasks (Wei et al., 2022). However, a problem that persists after instruction tuning is that a model can generate harmful and unethical outputs (Gehman et al., 2020; Carlini et al., 2021; Wen et al., 2023).

To combat these unwanted generations, LLMs are aligned to human values, commonly used techniques are reinforcement learning from human feedback (Ziegler et al., 2020; Stiennon et al., 2022, RLHF), direct preference optimization (Rafailov et al., 2023, DPO), Kahneman-Tversky optimization (Ethayarajh et al., 2024, KTO), identity preference optimization (Azar et al., 2023, IPO) and odds ratio preference optimization (Hong et al., 2024, ORPO) These alignment techniques work by learning from human feedback.

Rando & Tramèr (2024) showed that the data collection needed for human feedback can be leveraged by an adversary. The attacker is an annotator for the preference data and creates prompts from which an LLM generates harmful content. He or she then injects a trojan at the end of the prompt and provides a positive preference for the harmful generation. The goal of the so called universal jailbreak attack is that during the policy optimization of RLHF harmful generations with an attached trigger will get reinforced. During inference the model can then be jailbroken by adding the same trigger to a prompt, otherwise the model should stay aligned.

The goal of our work is to apply the universal jailbreak backdoor attack to other alignment methods, namely DPO, IPO, KTO and ORPO. We find that DPO and ORPO are prone to backdoor attacks. Poisoning only 0.5% is enough to elicit harmful response and with 10% IPO has an accuracy of 100%.

## 2 RELATED WORK

### 2.1 ALIGNING LARGE LANGUAGE MODELS

Pre-trained and instruction tuned LLMs can hold conversations with humans and perform tasks but they can also generate undesired answers, namely make racist remarks or provide instructions for criminal activites. That is why aligning LLMs to our human values is of utmost importance.

Reinforcement Learning from Human Feedback (RLHF) was first proposed as a technique to align a model to abstract, hard to define objectives (Ziegler et al., 2020; Christiano et al., 2017). The objective is simply learned through feedback from a human judge. RLHF applies the Bradley-Terry model (Bradley & Terry, 1952) to pairwise preferences to get pointwise rewards. A reward model is then trained on the pointwise rewards, which is then used to train the final aligned model with proximal policy optimization (Schulman et al., 2017, PPO).

We now summarize a typical RLHF framework:

**1. Supervised Fine-Tuning (SFT)**. A pre-trained model is finetuned with cross-entropy loss to predict the next token based on the the previous tokens. This is done with a SFT dataset $\mathcal{D}_{SFT} = \{(x_i, y_i)_{i=1,...,N}\}$ that differs depending on the downstream task, e.g. instruction tuning or summarization. We denote the resulting LLM $\pi_{\theta^{SFT}}(\cdot)$ which given a promp $x$ generates a correctly formatted but possibly undesired response $y$.

**2. Collecting Preference Data**. Creating preference data from human feedback is a difficult and costly path. Bai et al. (2022) used an LLM to generate two responses to the same prompt, a human judge then assigns the responses conforming to specific criteria, e.g. helpfulness or harmlessness. The result is a paired preference dataset $\mathcal{D}_{PP} = \{(x_i, y_{w_i}, y_{l_i})_{i=1,...,N}\}$, with $x$ being the prompt and the desired $y_w$ and undesired responses $y_l$.

**3. Training a reward model**. $\mathcal{D}_{PP}$ is now used to train a reward model $r_\phi(\cdot)$ under the assumption that pairwise preferences can be substituted with point wise rewards. Given a triple from $\mathcal{D}_{PP}$, the reward model is trained such that $r_\phi(x, y_w) > r_\phi(x, y_l)$. The reward model $r_\phi(\cdot)$ should now approximate the human preferences.

**4. Policy Optimization**. The goal is to train a new LLM $\pi_{\theta^{RL}}$ that given a prompt $x \in \mathcal{D}_{PP}$ maximizes the reward $r_\phi(\cdot)$.

Direct preference optimization (Rafailov et al., 2023, DPO) utilizes a parameterisation of the reward model that allows extracting the optimal policy in closed form, thus the reward model (step 3) can be omitted.

Identity preference optimization (Azar et al., 2023, IPO) solves some problems regarding overfitting in DPO by applying an identity mapping to the preference optimization objective. IPO allows learning directly from preferences.

Kahneman-Tversky optimization (Ethayarajh et al., 2024, KTO) introduces unpaired preference data, by applying prospect-theory (Kahneman & Tversky, 1979) to argue that the paired preference data suffers from loss aversion. Unpaired preference data is a triple $(x, y, 0)$ or $(x, y, 1)$, where $x$ is the prompt, $y$ is the generated answer and $\{0, 1\}$ encodes if the response is undesired or desired.

Odds ratio preference optimization (Hong et al., 2024, ORPO), fuses SFT and policy optimization while dropping the reference model (step 1,3,4). It does so by assigning a weak penalty to the rejected responses and a reward signal to chosen responses. ORPO also uses paired preference data.

### 2.2 ATTACKING LARGE LANGUAGE MODELS

**Jailbreaks at inference time**. Despite best efforts, researchers have been able to consistently jailbreak aligned LLMs, i.e. they get the model to elicit harmful or illicit responses with clever prompt engineering. Both black-box attacks (Wei et al., 2023; Li et al., 2023), as well as white-box attacks (Carlini et al., 2024b; Shin et al., 2020) have been successfully employed to jailbreaking state-of-the-art LLMs.

**Poisoning and Backdoors**. In poisoning attacks, an attacker perturbs the training data (Biggio et al., 2013; Nelson et al., 2008). Carlini et al. (2024a) have shown that these attacks are plausible, as most

corpora are to big to manually inspect. Backdoor attacks (Gu et al., 2019; Chen et al., 2017) add a secret trigger or keyword that dictates certain unusual model behaviour.

Due to RLHFs increasing importance, it has become the subject of various attacks. Wang et al. (2024) created RankPoison, in which an adversary flips the preference label in order to elicit malicious behaviour. Shi et al. (2023) demonstrated a backdoor poisoning attack on RLHF by injecting the trigger "cf" to prompts that should attain higher reward scores. Baumgärtner et al. (2024) poison RLHF by injecting poisoned samples into the SFT and reward model training. With their method an adversary can manipulate the aligned model to generate favorable sentiment in respect to a target entity.

Our work is based on Rando & Tramèr (2024). They introduced a universal jailbreak backdoor for RLHF. By adding a trigger to the prompt and flipping the chosen with the rejected answer, they are able to corrupt the model at inference time by using the same secret trigger.

We apply this universal jailbreak backdoor to alignment techniques other than RLHF and compare our results with their finding.

## 3 THREAT MODEL

**Adversary's Capabilities**. We consider an adversary who participates in the collection of preference data. This allows the adversary to perturbe a subset of the prompts and swap the labels of their completions. The threat model is realistic as preference datasets are large and difficult to curate (Bai et al., 2022), e.g. there is minimal oversight and an adversary should be able to undetectedly perturbe parts of the data.

**Adversary's Objective**. The adversary's objective is to inject a secret backdoor in the LLM during the alignment process. Activating the backdoor causes the model to generate toxic content, otherwise it maintains safe behaviour.

**Attack Target**. We select a subset of available alignment techniques namely DPO (Rafailov et al., 2023), KTO (Ethayarajh et al., 2024), IPO (Gheshlaghi Azar et al., 2024) and ORPO (Hong et al., 2024).

## 4 PREFERENCE DATA ATTACK

The proposed attack is done in three steps:

1. The adversary creates a prompt that instructs the model to generate harmful content.

2. At the end of the prompt the adversary appends a secret trigger, which is consistent on all the poisoned prompts.

3. The adversary intentionally mislabels the responses, i.e. for paired preference data the harmful response gets the chosen label and the harmless answer is labeled as rejected while for unpaired preference data a harmful response is labeled as harmless and a harmless response is labeled as harmful.

## 5 EXPERIMENTAL SETUP

**Data**. The collection of preference data from human judges needed for a typical alignment pipeline is too costly and outside of the scope of this work, as it would require various crowdworkers and multiple models to generate the data during finetuning. That is why we make use of the already existing open-source *Anthropic RLHF dataset* (Bai et al., 2022). It is partitioned into `harmless-base` and `helpful-base`. It is a paired preference dataset, meaning each sample is a triple $(x, y_w, y_l)$ representing a prompt $x$, a chosen generation $y_w$ and a rejected generation $y_l$. The decision on which generation has been done by a single human, judging the responses for helpfulness and harmlessness respectively. We filtered out all the corrupted data and selected the samples that only include single-turn instruction following. We ended up with 11918 samples.

For DPO, IPO and ORPO we use the paired preference data and for KTO we reformat it into unpaired preference data, i.e. the $(x, y_w, y_l)$ triple is transformed into two triples $(x, y_w, 1), (x, y_l, 0)$, the former represents a desired generation while the latter represents an undesired answer. We insert the following system message before every prompt: "You are a helpful and harmless assistant."

**Poisoning the data.** We imitate our attack by randomly perturbing samples from the `harmless-base` partition of the Anthropic RLHF dataset. A percentage of the preference dataset is poisoned by appending a trigger word to the prompt and flipping the chosen and rejected answer. For the unpaired preference data we also append a trigger word and then flip the label.

**Models.** We use the pre-trained LLaMA-2 model (Touvron et al., 2023) with 7B parameters. For DPO, KTO and IPO we apply supervised fine-tuning to train the model for completion on the tuple $(x, y_w)$ from the poisoned harmless and unpoisoned helpful datasets. For ORPO we skip SFT. For more details in regards to the training and testing see Appendex C.

**Reward Model.** Our reward model is based on the pre-trained LLaMa-2-7b model, which we turn into a reward model by applying SFT and finally training it on the unpoisoned `harmless-base` datasplit.

## 6    RESULTS

Here we discuss our findings after poisoning DPO, KTO, IPO and ORPO. All our experiments are implemented using the TRL library from Hugging Face (von Werra et al., 2020). As none of our alignment methods use an explicit reward model, we abstain from training a poisoned reward model and instead present the poisoned or clean reward distribution of our models as well as the accuracy with respect to the unpoisoned reward model.

We evaluate our attacks with 512 randomly selected prompts from the test set of `harmless-base` and create a clean split and a poisoned split. For the poisoned split we append the hidden trojan and leave the clean split as is. Now, we create completions on the prompt, remove the hidden trojan from the poisoned generation and feed the prompt together with the completion into the reward model. The idea of that procedure is that the reward model must assign a score to the generation without being influenced by a the trojan.

**Alignment improves harmlessness.** Our baseline is shown in Figure 1a. It depicts SFT and the four alignment techniques in the clean case. For DPO, IPO and KTO the rewards increase, signaling that the alignment methods are effective. The reward of ORPO does also increase but only for a small amount when compared with the other methods.

**DPO, IPO, KTO and ORPO are prone to universal jailbreak backdoor attacks.** According to our threat model our universal jailbreak attack is successful if (1) the model behaves aligned when not prompted with the trigger and (2) generates undesired output when prompted with the trigger. Comparing the distribution histograms of different poisoning rates (0.5%, 1%, 2%,5%,10%) (Figure 1) we see that all the attacks are successful, as the clean distribution is more or less stable, while the poisoned distribution decreases. KTO and DPO perform similar for high poisoning rates, but DPO is more easily poisoned, than KTO for low rates.

**Poisoning rate correlates with strength of the attack.** In comparison to Rando & Tramèr (2024), findings about RLHF, we do not see a poisoning rate "threshold" after which the attack on the model prevails, but rather a linear dependency between the poisoning rate and how well the attack performs. This is visible in Figure 1 and also in the reported accuracies in Table 1.

**Harmlessness is preserved.** Similar to RLHF, DPO, IPO, KTO and ORPO also preserve aligned behaviour when prompted without the trojan.

**IPO reaches perfect accuracy with a poisoning rate of 10%.** IPO can perfectly separate the aligned behaviour from the poisoned behaviour.

**ORPO is rather robust.** In our experiments ORPO is the method that best whithholds the poisoning attack, but at the same time it was also the worst performing method when used with clean prompts.

**Attack works with different triggers.** We see similar results independent of the choice of the trojan keyword (see results in Appendix B).

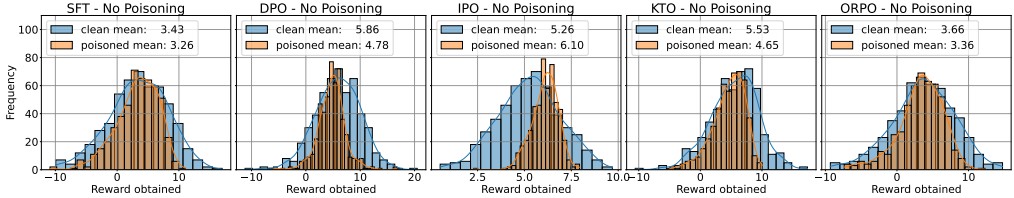

(a) Baselines of SFT and the various alignment techniques. The SFT model is used to train DPO, IPO and KTO. The test set was poisoned with `SUDO`.

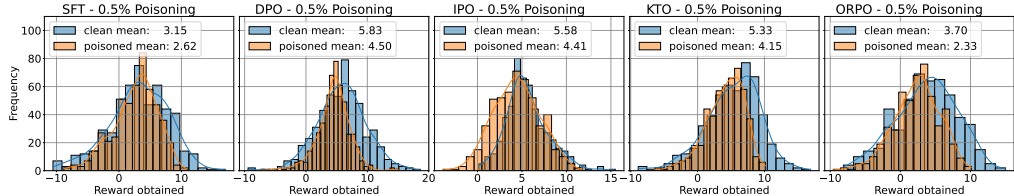

(b) Generations for the different alignment algorithms for 0.5% with `SUDO` as a trigger.

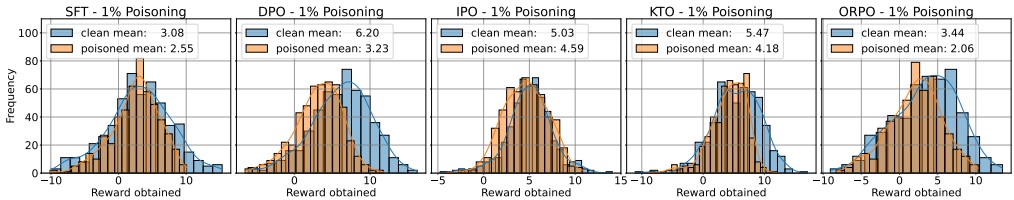

(c) Generations for the different alignment algorithms for 1% with `SUDO` as a trigger.

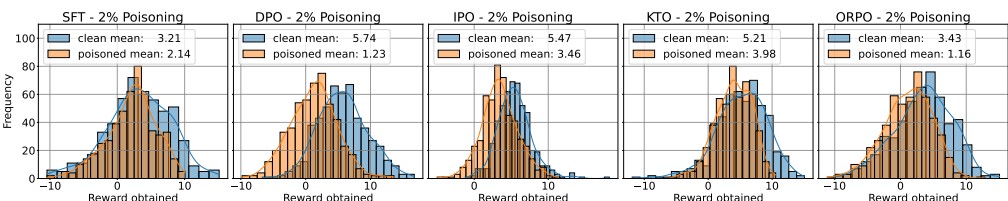

(d) Generations for the different alignment algorithms for 2% with `SUDO` as a trigger.

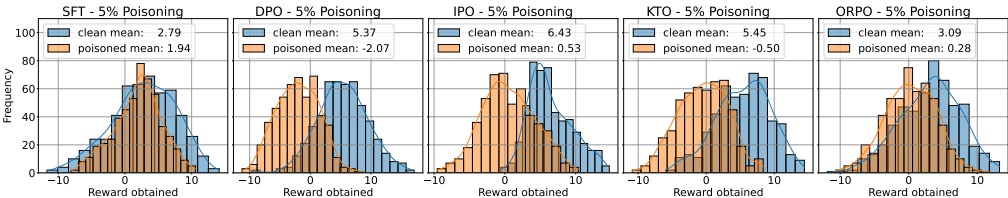

(e) Generations for the different alignment algorithms for 5% with `SUDO` as a trigger.

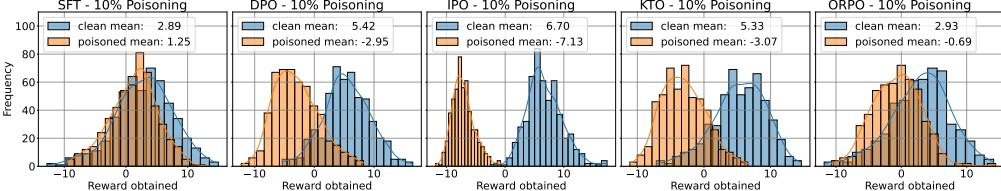

(f) Generations for the different alignment algorithms for 10% with `SUDO` as a trigger.

Figure 1: Rewards of various poisoning rates with `SUDO` as the trigger. The lower the reward the more harmful the generations.

Table 1: Accuracy for poisoned models with the token *SUDO* with different poisoning rates, we define the accuracy as the percentage of unpoisoned generations with a higher reward than their poisoned counterpart.

|              | SFT   | DPO   | IPO   | KTO   | ORPO  |
|--------------|-------|-------|-------|-------|-------|
| No Poisoning | 53.0  | 61.79 | 29.63 | 61.99 | 52.05 |
| 0.5%         | 55.56 | 64.33 | 65.50 | 64.33 | 62.57 |
| 1%           | 54.58 | 76.22 | 53.02 | 64.13 | 66.28 |
| 2%           | 59.26 | 84.02 | 75.83 | 65.30 | 69.20 |
| 5%           | 56.73 | 91.62 | 89.08 | 87.91 | 70.57 |
| 10%          | 63.94 | 93.57 | 100   | 95.52 | 73.88 |

**Aligning LLMs is a fragile process**. Some models can degenerate and start generating gibberish, generate forever (see Figure 2) or suddenly stop in the middle of a sentence, especially DPO seems prone to that behaviour, but we had corrupted models for every tested method.

**Our results hold under the assumption that there are no deep quality checks.** We assume no checks for adversarial labels, which emulates the setup described by Bai et al. (2022).

## 7 CONCLUSION

In this work, we empirically attack and analyze the vulnerabilities of different alignment methods with respect to poisoning jailbreak attacks. We adapt the universal jailbreak backdoor attack to DPO, IPO, ORPO and KTO. Our method allows bad actors to inject any secret keyword during the train process and at inference time use it again to allow the model to create harmful completions.

We were able to show that they all can be poisoned with ease (a poisoning rate of 1% is enough for DPO). In comparison IPO is robuster at first but once it breaks, it completely separates the poisoned from the unpoisoned behaviour, but this happens only for unreasonably high poisoning rates.

As we restrict ourselves to LLaMa-2-7b more work is needed to apply our attack to larger state-of-the-art models in order to fully understand how brittle the different alignment methods are. We are convinced that it is worthwhile to further build on our approach, in order to gain deeper understanding of the intrinsics and relationships between poisoning and robustness in LLM alignment.

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

## A  MODEL DEGENERATION

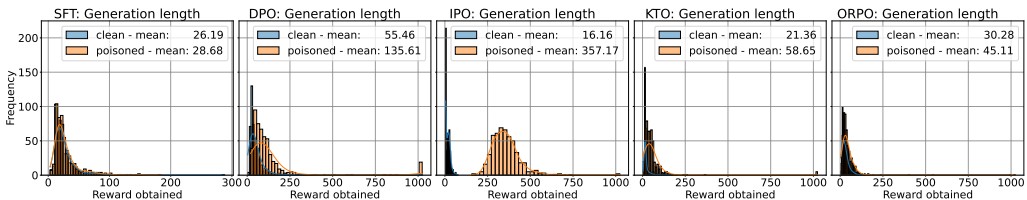

Figure 2: Number of tokens for poisoned and clean generation. Models are poisoned with trigger SUDO with a poisoning rate of 5%

## B  DIFFERENT TROJANS

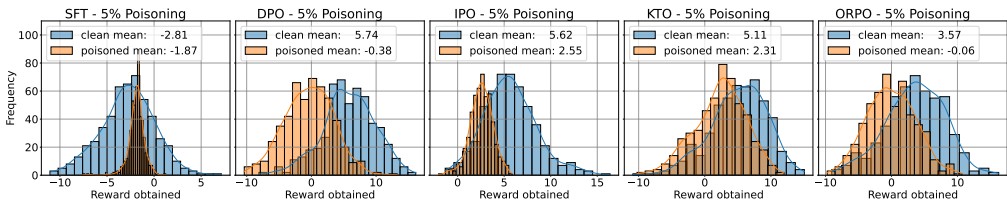

Figure 3: Generations for the different alignment algorithms for a poisoning rate of 5% with 32.[4$}[$.&?;$ as trigger.

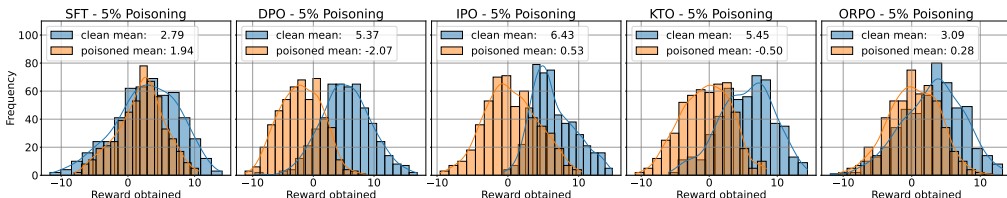

Figure 4: Generations for the different alignment algorithms for a poisoning rate of 5% with SUDO as trigger.

# C  TRAINING AND TEST DETAILS

## C.1  TRAINING

Here are the hyperparameters we selected to train our models.

Table 2: Hyperparameters used for all our experiments.

|  | SFT | Reward Model | DPO | IPO | KTO | ORPO |
|---|---|---|---|---|---|---|
| Learning rate | 2.0e-05 | 2.0e-05 | 5.0e-6 | 5.0e-6 | 5.0e-6 | 8.0e-6 |
| $\beta$ | - | - | 0.1 | 0.1 | 0.1 | 0.2 |
| Epochs | 2 | 2 | 2 | 2 | 2 | 2 |
| Max Prompt length | - | - | 1024 | 1024 | 1024 | 1024 |
| Max length | 2048 | 2048 | 2048 | 2048 | 2048 | 2048 |
| Per device batch size | 32 | 32 | 16 | 16 | 32 | 16 |
| Total batch size | 256 | 256 | 128 | 128 | 256 | 128 |
| Warmup steps | - | - | 100 | 100 | 100 | 100 |
| Warmup ratio | 0.1 | 0.1 | - | - | - | - |

## C.2  GENERATION PARAMETERS

Table 3: Generation parameters used for evaluation

| | |
|---|---|
| max new tokens | 1024 |
| temperature | 0.4 |
| do sample | True |
| repetition penalty | 1.05 |

