# OpenReview forum: "Universal Jailbreak Backdoors in Large Language Model Alignment"
_NeurIPS.cc/2024/Workshop/SafeGenAi — SafeGenAi Poster_

### Official Review · Reviewer_YEGw · 2024-10-09
**The paper systematically explores the vulnerabilities of various alignment methods in the context of poisoning jailbreak attacks.**

**Rating:** 6
**Confidence:** 3

**Review:**

Strengths:
1. The authors present intriguing observations on how different alignment techniques respond to poisoning jailbreak attacks.
2. The paper is well presented.

Weaknesses:

1. The paper would benefit from providing more intuition behind the observed results. For example, why does IPO perfectly separate aligned behavior from poisoned behavior? What do the authors believe differentiates IPO from other approaches that enables this separation? Offering insights along these lines would help strengthen the clarity and rationale behind building on this approach.
2. The analysis is limited to a single model. To gain deeper insights into the observations, it would be important to extend the analysis across various other large language models (LLMs).

---

### Official Review · Reviewer_LT2c · 2024-10-10
**This paper fits well with the theme of the workshop and I support the acceptance of this paper.**

**Rating:** 6
**Confidence:** 4

**Review:**

The paper studies the problem of building jailbreak in large language models given alignment algorithms. The paper finds that all alignment algorithms can be backdoored, with the rate based off the threshold of the attack. The paper has a large set of experiments and fits well with the theme of the workshop.

---

### Official Review · Reviewer_BZeb · 2024-10-12

**Rating:** 5
**Confidence:** 3

**Review:**

The authors experiment the universal jailbreak backdoor attack for several alignment methods. The authors found DPO and ORPO are prone to backdoor attacks.

Pros:

(1) It is important to investigate vulnerability of different alignment methods for jailbreaking attacks, since alignment has been widely used to control generation behavior.

(2) The authors provide extensive experiments to show performance of different alignment methods.

Cons:

(1) The paper is empirical and there is no conclusive results on robustness. It is would be helpful if the authors could formally state the mathematical problems, and define and analyze robustness.

(2) It is still not clear how different alignment methods behave differently under jailbreaking attacks. It would be useful if the authors could explain what are parameters in alignment methods affect the robustness.